# Environmental Effects during Early Life-History Stages and Seed Development on Seed Functional Traits of an Australian Native Legume Species

**DOI:** 10.3390/biology13030148

**Published:** 2024-02-27

**Authors:** Fernanda C. Beveridge, Alwyn Williams, Robyn Cave, Sundaravelpandian Kalaipandian, Mirza M. Haque, Steve W. Adkins

**Affiliations:** 1School of Agriculture and Food Sustainability, The University of Queensland, Gatton, QLD 4343, Australia; alwyn.williams@uq.edu.au (A.W.); r.cave@uq.edu.au (R.C.); s.kalaipandian@uq.edu.au (S.K.); mirzamobashwerul.haque@uq.net.au (M.M.H.); s.adkins@uq.edu.au (S.W.A.); 2Department of Bioengineering, Saveetha Institute of Medical and Technical Sciences (SIMATS), Saveetha School of Engineering, Chennai 602105, India

**Keywords:** climate change, moisture stress, physical dormancy, plant recruitment, seed ecology, seed functional traits

## Abstract

**Simple Summary:**

Seed-based restoration is likely to become less successful with climate change due to reduced seed germination and seedling emergence. Studying how native plant species respond to environmental stress using a whole life cycle approach can help predict climate change impacts on successful plant establishment. Therefore, the aim of this study was to understand how temperature and moisture stress affect seed germination and seedling emergence together with determining how soil moisture stress can impact the reproductive biology of plants. A native Australian legume species, *Desmodium brachypodum* A. Gray, was selected as a model species. The seed germination and seeding emergence were reduced by more than half when exposed to elevated temperatures and moisture stress. When the plants produced seeds under moisture stress, the duration of seed production and the number of produced seeds decreased. On the other hand, no differences were observed in the seed traits of those produced seeds. In conclusion, the reproductive output of *D. brachypodum* had low seed variability under moisture stress, which might be useful when sourcing seeds from climates with high variability. Even so, a reduction in the seed quantity under maternal moisture stress can impact the long-term survival of restored plant populations.

**Abstract:**

Understanding how seed functional traits interact with environmental factors to determine seedling recruitment is critical to assess the impact of climate change on ecosystem restoration. This study focused on the effects of environmental factors on the mother plant during early plant life history stages and during seed development. *Desmodium brachypodum* A. Gray (large tick trefoil, Fabaceae) was used as a model species. Firstly, this study analyzed seed germination traits in response to temperature and moisture stress. Secondly, it investigated how seed burial depth interacts with temperature and soil moisture to influence seedling emergence traits. Finally, it determined if contrasting levels of post-anthesis soil moisture could result in changes in *D. brachypodum* reproductive biology and seed and seedling functional traits. The results showed that elevated temperature and moisture stress interacted to significantly reduce the seed germination and seedling emergence (each by >50%), while the seed burial improved the seedling emergence. Post-anthesis soil moisture stress negatively impacted the plant traits, reducing the duration of the reproductive phenology stage (by 9 days) and seed production (by almost 50%). Unexpectedly, soil moisture stress did not affect most seed or seedling traits. In conclusion, elevated temperatures combined with low soil moisture caused significant declines in seed germination and seedling emergence. On the other hand, the reproductive output of *D. brachypodum* had low seed variability under soil moisture stress, which might be useful when sourcing seeds from climates with high variability. Even so, a reduction in seed quantity under maternal moisture stress can impact the long-term survival of restored plant populations.

## 1. Introduction

Successful plant recruitment is critical for the long-term survival of sexually reproducing plants [1]. However, early-life history plant stages are a major bottleneck in natural and restored ecosystems, with low seedling emergence often limiting plant recruitment [2,3]. Research has shown that seedling losses occurring between germination and emergence can be as high as 90% [4], with the most vulnerable stage being the transition from the germinated seed to the emerged seedling [5].

Plant recruitment stages are highly vulnerable to edaphoclimatic factors, particularly ambient temperature and soil moisture stress. Temperature is a major influence on plant recruitment from seeds, synchronizing seed germination and seedling emergence to occur when the environmental conditions are optimal for seedling establishment [6]. Temperature also influences seedling growth [7] and mortality [8]. Across ecosystems, limited soil moisture content can reduce seedling emergence [9]. To predict the plant responses to temperature and soil moisture changes, it is crucial to investigate how seed functional traits interact with environmental conditions to define a species’ realized niche [10,11].

Interactions between certain seed functional traits and environmental factors determine the germination probability, with different seed functional traits (such as seed mass, dormancy, and germination requirements) modulating germination timing and seedling fitness [11]. Seed mass is an important functional trait relating to germination and is used for the prediction of plant community regeneration patterns [12]. Seed size is correlated with improved seedling establishment and survival [13] and has also been correlated with seedling survival under conditions of soil moisture stress [14]. The seedling emergence rate can influence establishment success and subsequent plant fitness. Even short delays in the time from seed germination to seedling emergence can result in significant differences in the final biomass and reproduction [15]. Therefore, early emergence is a strong fitness determinant that can increase reproductive success, especially in environments with strong resource competition.

Under conditions of environmental stress, dormancy is a critical trait modulating germination [16] and seedling survival [17]. Physical dormancy (PY) is found in several native Australian Fabaceae species [18]. Physical dormancy is controlled either by the seed testa or the endocarp and is a heritable trait [19]. Dormancy loss in seeds with PY occurs due to an irreversible structural change to the “water gap”, a specialized region of the testa [20]. In nature, PY loss is mainly modulated by the interaction of soil moisture and temperature, including events such as fires, dry or wet heat, and temperature fluctuations [21]. Different traits act together to determine the dormancy loss, including the percentage of PY seeds at dispersal from the mother plant, the environmental conditions needed to break PY, the time taken for PY loss to occur under certain conditions, and changes in the conditions needed to break PY over time.

The maternal environment during seed formation plays a crucial role in regulating the processes of seed development and, consequently, early plant life history [22]. Abiotic factors, mainly temperature and soil moisture availability, play an important role in determining certain seed functional traits, such as dormancy and germinability [23]. For seeds with PY, the seed coat becomes impermeable to water at seed maturity [24] due to the dehydration of the testa epidermal cells, with the level of PY depending on the moisture content reached by the seed during maturation drying [25]. Soil moisture conditions during seed development can influence the onset and depth of PY, ranging from all seeds having an impermeable seed coat to some or no seeds being impermeable to water [26]. Higher levels of moisture stress have been associated with higher levels of seed PY compared with seeds produced in years with high rainfall [25,27].

Climate change is already causing warming temperatures and rainfall patterns to become more unpredictable, with several areas of the globe becoming prone to elevated temperatures and drought [28]. Besides increased warming, extreme climatic events (heat waves, fires, and droughts) are predicted to increase in intensity and severity in Australia [29]. By altering the environmental cues sensed by seeds, climate change can compromise seedling emergence, establishment, and survival. Even small changes in certain climatic factors can produce significant changes in seed and seedling behavior [6]. Temperature increases in the Australian spring and summer are projected to be higher than those seen in autumn and winter [30]. This might severely impact seed germination in subtropical and tropical native Australian species that usually germinate at the beginning of the rainy season in spring/early summer.

Assessing how certain seed functional traits (such as seed mass, germination temperature, and moisture thresholds) interact with environmental factors (e.g., temperature and soil moisture) to determine seedling recruitment (seed germination, seedling emergence, survival, and growth) [31], is critical to assess the impact of climate change on the resilience of natural ecosystems. This study investigated how temperature and soil moisture interact to influence the early plant life history stages and seed development of the native Australian *Desmodium brachypodum* A. Gray plant (large tick trefoil, Fabaceae). This species was selected as a model species, given it has seeds with PY and is naturally found in locations predicted to experience increased temperatures and reduced soil moisture [32]. The objectives were to (1) analyze the seed germination traits (final germination percentage, mean germination time [MGT], and synchrony) in response to temperature and moisture stress (experiment 1); (2) investigate how seed burial depth interacts with temperature and soil moisture to influence seedling emergence traits (experiment 2); and (3) determine if different levels of post-anthesis soil moisture result in changes in the reproductive biology of *D. brachypodum*, particularly with phenological development (days to seed pod production and harvest) and seed functional traits (PY, germination rate and percentage, seed mass, seed longevity, and seedling traits) (experiment 3).

## 2. Materials and Methods

### 2.1. Study Species and Seed Source

*Desmodium brachypodum* is a widespread species in eastern Australia (Figure 1a), inhabiting inland tropical and subtropical environments [33]. It grows in dry forests, woodlands, and grassy areas, often on damp sites, and has several restoration benefits (such as fast growth, atmospheric nitrogen fixation, and is a food resource for native fauna). Flowering occurs from spring to autumn (Figure 1b). Seed pods are transversely joint, breaking at maturity into separate one-seeded sections that are individually dispersed. The seed is small and light, semi-spheroid in shape, and has a smooth, thick yellow testa with a small white elaiosome (Figure 1c).

The seeds were hand-harvested by Biobank Seeds (Uralla, New South Wales [NSW]) from a site near Narrabri, NSW (30°19′56.6616″ S, 149°46′52.464″ E) on 19 March 2020. The mature seeds from >100 individual plants were collected. After they were received, the seeds were transferred to a specialized seed store maintained at 15 ± 2 °C and 15 ± 5% relative humidity (RH) until used (30 days later). The average 100-seed weight was 27.0 ± 0.0 g, and the seed fill was 99 ± 1%.

### 2.2. Interactions between Temperature and Moisture Stress on Seed Germination (Experiment 1)

Prior to the experimental setup, the seeds were surface-sterilized in a 2% (*v*/*v*) sodium hypochlorite (NaOCl) solution for 10 min with two drops of Tween 20 surfactant (Labchem, Zelienople, PA, USA). The seeds were then washed three times with sterile deionized water and blotted dry. To alleviate PY, the seeds were soaked in hot water (95 ± 2 °C) for 2 min. The seeds were then exposed to four water potential levels of 0.0 MPa (control), −0.4 MPa (moderate stress), −0.8 MPa (severe stress), and −1.5 MPa (extreme stress). The target water potential values were achieved using different concentrations of polyethylene glycol (PEG) BioUltra 8000 (Sigma-Aldrich, 3050 Spruce Street, St. Louis, MO, USA) with sterile deionized water as the control treatment. The PEG solutions were prepared according to the protocol and equation of [35]:
[PEG] = [4 − (5.16ψT − 560ψ + 16)^0.5^]/(2.58T − 280)
where:

Temperature (T) is expressed in degrees Celsius (°C)

Osmotic potential (Ψ) is expressed in bar (1 bar = 0.1 MPa)

PEG amount, [PEG], is expressed as g of PEG g^−1^ of H_2_O

This equation is the result of a model that considers ψ varying quadratically with concentration and linearly with the temperature of the solution (the PEG solutions were separately prepared for each temperature). Each treatment had four replicates of 20 seeds that were placed in plastic Petri dishes (with a 9-cm diameter) containing two Whatman No. 1 filter papers moistened with 5 mL of sterile deionized water (control) or PEG solution. The Petri dishes were closed and placed into a transparent plastic box (40 × 20 × 10 cm; l/w/d) lined with three layers of saturated paper towels, then closed with an airtight lid to reduce the evaporation. Each box containing a complete set of treatments was placed in an incubator (TRIL-750 Illuminated Refrigerated Incubator, Thermoline, Wetherill Park, Australia) at either 25/15 ± 1 °C (day/night; typical Spring condition) or 30/20 ± 1 °C (day/night; climate change altered Spring condition), with a 12/12-h day/night photoperiod and thermoperiod.

A completely randomized design was used. The seeds were checked for germination three times per week. Germination was defined as a radicle protrusion of 1 to 2 mm. All the germinated seeds were removed to avoid interaction with the ungerminated seeds. At the end of the experiment (28 days), a cut test was used to evaluate the ungerminated seed viability. The seeds with white, turgid embryos were considered alive, while the brown, flaccid embryos were considered dead. The MGT (the reciprocal of the rate of germination) and germination synchrony (Appendix A) were calculated using the GerminaR package [36] in the R statistical software environment version 4.2.2 [37].

### 2.3. Seed and Seedling Performance following Burial, Elevated Temperature, and Reduced Soil Moisture (Experiment 2)

#### 2.3.1. Soil Moisture

The topsoil was sourced from agricultural land at the UQ Gatton Campus farm (5391 Warrego Highway, Gatton 4343, Qld). The soil was air dried for 7 days, then homogenized to become a 5 mm particulate diameter. To determine the moisture treatments, the plant’s available water content and the moisture retention curve of the black vertosol soil were determined using the pressure plate method [38] (Appendix B). Two soil moisture conditions were used (based on the preliminary experimental results): 100% plant-available water content (PAWC) (wet) and 60% PAWC (dry). For the 100% PAWC, the pots were wetted to 100% PAWC (a soil water content of 287 mL of water kg^−1^ of soil), then allowed to dry until 90% PAWC (a soil water content of 258 mL kg^−1^) before being re-wetted again to 100% PAWC. For the 60% PAWC, the pots were wetted to 60% PAWC (172 mL kg^−1^) and then allowed to dry to 50% PAWC (144 mL kg^−1^) before being re-wetted to 60% PAWC.

#### 2.3.2. Seedling Emergence Trial

The seedling trial was conducted using non-dormant seeds (as described in experiment 1). Prior to sowing, black plastic ANOVApot^®^ pots (diameter: 140 mm) were filled with soil to a uniform weight (of 1.00 kg of dried soil per pot). The soil was wetted to field capacity (−0.01 MPa) and allowed to dry undercover at an ambient temperature until the desired moisture was reached (100 or 60% PAWC). Each pot was divided into halves, where 25 seeds were sown on the soil surface (in one half of the pot), and 25 seeds were buried at 2 cm depth (in the other half of the pot). Four pots were used per treatment, with 16 pots total. The pots were placed in growth chambers (A2000 Conviron) at either 28/15 °C (simulated spring/early summer conditions) or 33/20 °C (simulated spring/early summer climate change conditions) with a 14/10-h day/night photoperiod and thermoperiod, and ca. 60% RH. The pots were weighed and watered every two days to maintain the soil at the required soil moisture. At 30 days post-sowing, all the emerged seedlings were harvested (to avoid seedling competition and to prevent the emerged seedlings from inhibiting further seedling germination). The pots were monitored twice weekly for another 30 days for further emergence. The following traits were measured: seedling emergence, defined as the protrusion of the fully expanded cotyledons above the soil substrate; seedling development, measured as the day of emergence and the day of the first fully expanded leaf; and seedling survival, measured as the live seedlings at the end of the trial (60 days post sowing) minus the total seedlings that emerged. The non-emerged, buried, and surface-sown seeds were removed from the experiment and categorized as either (1) germinated but failed to emerge/establish or (2) ungerminated.

### 2.4. Effects of Post-Anthesis Maternal Soil Moisture Stress on Seed and Seedling Traits (Experiment 3)

#### 2.4.1. Parental Seed Germination and Plant Growth

Four replicates of 50 non-dormant seeds were placed into Petri dishes to germinate, then placed in an incubator set at optimum temperature and light conditions (as described in experiment 1). Six days after germination, all the seedlings were transplanted into multiple-celled trays (individual cells 4.0 × 4.0 × 8.5 cm; w/l/h) containing Osmocote^®^ native premium potting mix, composed of plant mulch, topsoil, wood dust, manure, mushroom material, controlled-release fertilizer, wetting agent, acidic pH, trace elements, low phosphorus, iron, and magnesium. The seedling trays were kept under ambient glasshouse conditions (evaporatively cooled, with float glass with thermal screens to reduce heat load, temperature range: ±6 °C) with natural illumination (14/10 h light/dark) and sprinkler irrigation.

Ninety-six days after seedling planting, the healthy and uniformly-sized seedlings were transplanted into plastic pots (diameter: 250 mm; one seedling per pot) containing 7 kg of soil (the same topsoil as experiment 2). The pots were transferred to a temperature-controlled glasshouse (toughened glass with thermal screens to reduce thermal heat load, chilled water facility, temperature control ± 2 °C). The temperature regime was set based on a springtime condition when the seedlings would start growing in their natural habitat and were selected based on 10-year average temperatures where the species commonly occur (Appendix C). Controlled release Osmocote^®^ Pro Low P Native fertilizer was applied 8 days after transplanting, and liquid fertilizer (Powerfeed^®^, macronutrients, fish, and natural soil conditioners) was applied monthly for the first 3 months. The plants were hand-watered to maintain 100% PAWC until anthesis.

#### 2.4.2. Soil Moisture

Post-anthesis (after the first flower opened and anthers dehisced—162 days after sowing), the pots were randomly assigned to three soil moisture levels. Based on the calculated PAWC, the pots were maintained at 287 mL water kg^−1^ soil for 100% PAWC (wet soil), 201 mL water kg^−1^ for 70% PAWC (mild soil moisture stress), or 115 mL water kg^−1^ for 40% PAWC (moderate soil moisture stress). The pots were weighed several times a week and watered accordingly to maintain the desired soil moisture.

#### 2.4.3. Experimental Design and Data Collection

All the pots were randomly distributed on benches, creating three blocks (replicates). Each block contained seven pots per treatment, providing a total of 21 pots per block. The plant development stages (days to anthesis [days from seedling emergence to anther dehiscence], days to first seed pod [days from anthesis to the first fully formed seed pod], and days to seed maturation [days from first seed pod formation to physiologically mature seeds]) were recorded, and post-anthesis, the following measurements were taken weekly: plant height (from the soil surface to the tip of the tallest plant part), number of open flowers (counted once the flower buds were fully open), seed production (assessed as the number of seeds produced per seed pod per plant) and seed physiological maturity (determined by their change in color from green to dark brown, with the seed at the tip of the pod being the first to change color, then lastly the seed attached closest to the peduncle) for a total of 98 days. At the end of the experiment, the above-ground dry biomass was determined via oven drying at 65 °C for 48 h.

#### 2.4.4. Seed Harvesting

The mature seed pods were harvested every 2 days for a period of 73 days. The seeds were removed from the pods, air-dried, and stored in a specialized seed store until they were used, which was 7 days after the last harvest. The number of seeds produced per plant was determined, then, six replicates of 200 randomly selected seeds per block per treatment were used to measure the mean 100-seed weight. The seed fill was also examined (as in experiment 1).

#### 2.4.5. Germination Test of the Progeny Seeds

The seeds were bulked together into three lots based on the soil moisture stress treatment and tested for germination under optimum temperature and light conditions (following the protocol in experiment 1). For each experiment, four replicate Petri dishes, each containing 25 seeds from the respective soil moisture treatments, were used.

#### 2.4.6. Controlled Aging Test (CAT)

A CAT test was performed to measure the effects of the post-anthesis soil moisture treatment on the seed longevity. The seeds were placed in open glass vials inside a sealed box with 47.0 ± 1.5% RH at 20 ± 1 °C for 6 days to pre-equilibrate and to ensure minimal seed moisture changes occurred at the start of the aging treatment (LiCl solutions were used to create the desired RH conditions). Then, the seeds were transferred to the CAT environment, where the glass vials were placed inside a sealed box with 60.0 ± 1.5% RH and put in an oven at 45.0 ± 0.5 °C under darkness [39]. The temperature and RH were recorded (Temperature Technology, Adelaide, Australia), and water was added whenever the RH fell by ca. 1%. One sample of 60 seeds was removed from the aging environment after 1, 4, 11, 28, 65, 94, and 129 days for germination testing (the days were determined based on a preliminary experiment with *D. brachypodum*), then immediately placed into a germination test as four replicates of 15 seeds [40].

#### 2.4.7. Maternal Effects on Progeny Seedling Performance

To determine the post-anthesis soil moisture effects on the progeny seedling traits, 14 days after sowing, the healthy and uniform seedlings from the germination trial were transplanted to multiple celled trays (4.0 × 4.0 × 8.5 cm; w/l/h) filled with topsoil (the same as experiment 2). The trays were randomly distributed onto a bench inside a growth chamber set at a 25/15 °C thermoperiod and matching a 12/12-h photoperiod (the same incubator as experiment 1). The seedlings were thinned down to 40 per mother plant moisture treatment. After 28 days, the seedling height was measured and the number of leaves per plant were counted, after which they were harvested, and the above-ground dry biomass was determined.

### 2.5. Statistical Analysis

The seed germination percentages for experiments 1 and 3 were analyzed using generalized linear models (GLMs) fitted with a logistic link function and a binomial error structure to determine the factorial effects using the glm function in R. A fully factorial model including all factors and interactions was computed. The final germination was analyzed as the response variable, with temperature and moisture as the explanatory variables for experiment 1, and moisture as the explanatory value for experiment 3. The MGT and germination synchrony were analyzed using linear models for experiments 1 and 3. A Tukey post-hoc test was carried out for the mean comparison between the species for each parameter.

For experiment 2, the seedling emergence was analyzed using generalized linear mixed effects models (GLMMs), with a nested Poisson regression model used for the final emergence percentage, seedling development, and survival. For the seedling survival, only the temperature and moisture were used as the explanatory variables, as it was assumed that the seed burial would not affect the final seedling survival [41]. A Tukey post-hoc test was carried out for the mean comparison between the species for each parameter. For experiment 3, the effects of soil moisture on the plant morphological traits (plant height and dry biomass), seed production, 100-seed weight, and plant phenological stages were analyzed as an RCBD (with the soil moisture and block as the factors). A Fisher’s least significant difference (LDS) test was used for the pairwise comparison between the means. The seedling traits were analyzed using a one-way ANOVA, with moisture as the explanatory variable.

To estimate the seed longevity, the time taken for the viability (measured as the germination) to decline by 50% under aging conditions (*P_50_*) was calculated [42]. The *P_50_* was determined using a Probit analysis and used as a measure of the relative seed longevity. This was achieved by fitting a GLM with a binomial regression and a probit-link function to the data (the number of seeds germinated relative to the number of seeds sown). The aging time was the explanatory variable, and the maternal soil moisture treatment was the fixed variable. Then, the *P_50_* for each soil moisture treatment was estimated using the *dose.p* function in the MASS package.

## 3. Results

### 3.1. Interaction between Temperature and Moisture Stress on Seed Germination (Experiment 1)

A significant interaction was observed between temperature and moisture stress on the germination traits of *D. brachypodum* (*p* < 0.01, Figure 2). At 30/20 °C, the germination was >96% from 0 to −0.8 MPa, then decreased significantly to 39.1 ± 6.4% at −1.5 MPa (Figure 2a). At 25/15 °C, the germination was the highest for the control and −0.4 MPa (>96%), then decreased significantly (*p* < 0.01) to 37.8 ± 22.1% for −0.8 MPa and 10.0 ± 7.1% for −1.5 MPa (Figure 2a). The mean germination time increased significantly (*p* < 0.01) with increasing moisture stress for both temperatures (doubling at 20/15 °C and increasing sixfold at 30/20 °C) (Figure 2b). Additionally, the MGT was significantly shorter (*p* < 0.01) for the seeds sown at 30/20 °C and down to −0.8 MPa (Figure 2b). The germination synchrony did not vary significantly (*p* > 0.05) from the control for most moisture stress conditions, with no trends observed for either temperature treatment (Figure 2c). Given the low germination percentage achieved by the seeds exposed to −1.5 MPa at 25/15 °C (10%), the MGT and germination synchrony were not calculated.

### 3.2. Seed and Seedling Performance following Burial, Elevated Temperature, and Reduced Soil Moisture (Experiment 2)

There was a significant interaction between the temperature and soil moisture treatment on the final emergence percentage (Figure 3a), but no treatment effects were observed for seedling development or survival (Figure 3b,c). The seeds exposed to 28/15 °C had a higher germination value than at 33/20 °C for both soil moisture treatments (*p* < 0.02). The highest-achieved emergence was for the buried seeds at 28/15 °C with 85.0 ± 8.7% emergence for 100% PAWC and 54.4 ± 15.4% for 60% PAWC. In contrast, the lowest emergence observed was for the seeds exposed to 60% PAWC and 33/20 °C (≤11% either buried or surface-sown). Additionally, within the same temperature treatment, the seeds exposed to 60% PAWC had lower germination than the seeds exposed to 100% PAWC (*p* < 0.01). The buried seeds had a higher seedling emergence percentage as compared to the surface-sown seeds at 28/15 °C for each soil moisture treatment (*p* ≤ 0.01), and it was observed that the buried seeds emerged and established earlier (Figure 4). On average, the seedlings developed the first leaf 3 days after emergence (Figure 3b), and there was 90% survival (Figure 3c). Most non-germinated seeds were dead.

### 3.3. Post-Anthesis Maternal Soil Moisture Stress Effects on Plant Traits, Seed Functional Traits, and Seedling Traits (Experiment 3)

Anthesis commenced ca. 194 days after parental seedling emergence (*ca.* 70 days after transplant to the glasshouse) and lasted for ca. 78 days. The days from anthesis to the first seed pod production did not differ between 100% PAWC and 70% PAWC (24 days on average), but for the 40% PAWC, the number of days to the first seed pod was lower (*p* < 0.02) (15 ± 2 days) than for 100% PAWC (Figure 5a). The seed pod production lasted for ca. 60 days. The seed harvesting started 191 days after parental seedling emergence and 96 days post-seedling transplant. The days from the first seed pod produced to the last harvested were not significantly different between the soil moisture treatments (ranging from 30 to 39 days).

The post-anthesis soil moisture had a significant effect on the maternal plant height, biomass, and number of seeds produced plant^−1^ (*p* = 0.03, *p* < 0.01 and *p* < 0.01, respectively; Figure 6). The plant height was 69 ± 3 cm for 100% PAWC, significantly taller than for 70 and 40% PAWC, which had plant heights of 60 ± 2 and 61 ± 3 cm, respectively (Figure 6a). The above-ground dry biomass was greatest for 100% PAWC, with 3.2 ± 0.2 g, as compared to 2.3 ± 0.2 g for 70% PAWC and 1.6 ± 0.2 g for 40% PAWC (Figure 6b). Moreover, the above-ground dry biomass at 70% PAWC was significantly greater than at 40% PAWC. There was a significant difference (*p* = 0.01) between the number of seeds produced in 100% PAWC (137 ± 20 seeds) and 40% PAWC (72 ± 10 seeds), but no significant difference (*p* = 0.06) was observed between 100 and 70% PAWC (95 ± 16 seeds) or between 70 and 40% PAWC (Figure 6c). The results showed that the plants produced similar numbers of seeds per seed pod, irrespective of soil moisture treatment (Figure 5b). Additionally, the seed mass did not vary across the different soil moisture treatments, with a 100-seed mass of 0.3 g. All soil moisture treatments produced 100% seed fill.

The seeds from all maternal soil moisture stress treatments had a high proportion of PY, as there was little (≤2%) or no seed germination for the control for all the soil moisture treatments (Figure 7). There was no significant difference in the final seed germination for the different soil moisture treatments (*p* = 0.6; Figure 7), with seeds treated with hot water achieving 78 ± 7% (100% PAWC), 81 ± 3% (70% PAWC) and 74 ± 4% (40% PAWC). The CAT test results showed that the maternal soil moisture treatments did not change the predicted longevity of the seeds in the soil seed bank, as the estimated *P_50_* for all the seeds corresponded to long-lived seeds (ranging from 483 days for 100% PAWC to 130 days for 40% PAWC). There were no significant differences (*p* > 0.06) between the soil moisture treatments for seedling height (1.6 cm on average), dry biomass (7.0 g on average), or leaf number (three leaves per seedling on average).

## 4. Discussion

In this study, we investigated how interactions between temperature and moisture impacted the early-life history stages of *D. brachypodum*, and we assessed how post-anthesis soil moisture affects the plants’ reproductive biology. This study shows that interactions between temperature and moisture significantly reduced seed germination and the seedling emergence percentage, while soil burial could improve the final emergence. Additionally, post-anthesis soil moisture stress had significant negative impacts on plant traits and reduced the duration of the phenology stage and seed production. Unexpectedly, moisture stress did not affect most seed or seedling traits, as was observed for other PY species. Below, we discuss these results and review the implications of future climate change on plant life history stages within a seed-based restoration context.

### 4.1. Temperature and Moisture Effects on Early Plant Life History Stages

The interaction between temperature and moisture stress impacted both the seed germination and seedling emergence traits. Moisture stress sensitivity is known to be dependent on germination temperature, where the inhibitory effect of restricted water availability for germination was observed as more pronounced at higher temperatures [43]. Nonetheless, the opposite effect was observed in this study. The seeds that germinated at higher temperatures (30/20 °C) had higher germination under moisture stress as compared to seeds that germinated at lower temperatures (25/15 °C). These results suggest that the higher temperatures might have been closer to the temperatures that *D. brachypodum* seeds are exposed to in their natural environment (subtropical and tropical eastern Australia), with seeds performing better under stress at their optimum germination temperature. Higher temperatures have been reported to hasten germination by speeding up the rate of chemical reactions within the seeds that result in germination [44].

Moisture stress significantly inhibited the germination and increased the MGT for both temperature treatments. Seeds kept under low water potentials do not absorb the minimum amount of water required to commence germination [45]. This could be attributed to seeds having deficient energy to trigger the germination process, given that after imbibition, the upregulation of respiratory pathway promotors is responsible for energy production, and water imbibition occurs much slower in the presence of low water potentials [43]. Additionally, increases in the MGT under moisture stress were previously reported in the literature for different species [43]. Even so, it was interesting to observe up to 29% germination at such low water potentials (at −1.5 MPa and 30/20 °C), which is equivalent to very dry soil conditions. This behavior suggests that *D. brachypodum* might have a low base water potential threshold for germination, which could be detrimental if germination occurs under suboptimal moisture conditions and the resulting seedlings are exposed to unfavorable emergence environments.

The interaction between higher temperatures and soil moisture stress hindered the seedling emergence percentage. High temperatures can exacerbate soil moisture stress through the evapotranspiration process. When high temperatures are combined with soil moisture stress, it can result in higher seedling mortality [46]. Additionally, there was higher emergence at 28/15 °C compared to 33/18 °C, irrespective of the soil moisture treatment. This information, coupled with the germination data, suggests that the optimum germination temperature for *D. brachypodum* is around 30 °C and that an increase of 3 °C could be detrimental to germination and emergence success. Given that increasing warming is projected to occur across Australia [47], with inland areas subject to a higher increase [28], plant recruitment from seeds might severely impact this species [33]. The seedlings that did emerge under temperature and soil moisture stress were able to develop into normal seedlings and establish successfully with high seedling survival for all the treatments. These results are consistent with other studies that show that the soil moisture stress experienced by seeds during germination does not always lead to lower seedling growth [43].

Seed burial improved the emergence percentages when the seeds were sown at 28/15 °C. Given the small and light nature of *D. brachypodum* seeds, it was expected that they would perform better when the surface was sown. It is well known that smaller seeds generally emerge better near or at the soil surface due to a lack of energy reserves to successfully reach the soil surface when buried [48]. Additionally, visual observation suggested that the buried seeds germinated, emerged, and established earlier than the surface-sown seeds. This could be explained by the fact that the seeds were lightly buried, which might have provided them with a favorable microenvironment for water uptake by having higher seed-to-soil (and hence moisture) contact, facilitating seed water imbibition and, therefore, germination.

Similar to our findings, Wang, et al. [49] observed that a seed burial of 2 mm improved the emergence and survival of small *Anabasis aphylla* L. (Amaranthaceae) seeds, where lightly buried seeds were better protected from drought stress. Additionally, small hard-seeded legume species (such as *Acacia* spp.) from southwestern Western Australia germinated better under darkness, which was associated with the ability of seeds to sense burial [50]. Germination at the soil surface might be more problematic for smaller seeds in seasonally dry environments. Smaller seeds possess relatively smaller amounts of internal water as compared to larger seeds after imbibition, which may not be enough to develop seedlings that are capable of overcoming the faster soil drying conditions occurring early in the rainy season [50]. Further experiments could investigate burying seeds at different soil depths to identify the optimal burial range under soil moisture stress and identify up to what burial depth the trade-off between moisture access versus energy reserves needed for emergence occurs.

### 4.2. Post-Anthesis Soil Moisture Effects on Plant Life History Traits

Soil moisture stress had a significant effect on the plant life history traits and affected the reproductive phenology of *D. brachypodum* by shortening the days from anthesis to seed pod production. This behavior is consistent with previous studies, where it has been observed that warming temperature and moisture stress can affect plant phenology stages [51]. Reduced soil moisture advanced and shortened the phenology stages for European high-elevation alpine species [51], and earlier flowering was observed in natural populations of *Brassica rapa* L. (field mustard, Brassicaceae), as an evolutionary shift to escape drought [52]. When soil moisture availability is limited, shifting to rapid flower development and maturity is an advantage that can allow reproductive success to be maintained. The mechanisms associated with this response include high transpiration and inefficient water use, enabled by high stomatal conductance, which leads to faster development. This maximizes carbon gain at the expense of transpirational water loss.

Maternal plant growth conditions are important in determining the seed size versus number trade-off by restraining the ability of plants to invest optimally in reproduction [44]. The results of the present study show that *D. brachypodum* responded to soil moisture stress by decreasing the seed number rather than altering the seed mass. The seed yield was also significantly affected, but no differences were observed in the seed mass, seed fill percentage, or the number of seeds produced per seed pod. Abiotic stress during seed production has been shown to have different effects on seed morphology. Wang, et al. [53] observed that seeds produced under salinity stress were larger than those produced under optimal conditions when studying the annual halophyte *Atriplex centralasiatica* (Popov) G.L.Chu (Amaranthaceae). Additionally, longer exposure to moisture stress significantly influenced the seed dimensions for 43 native south-eastern Australian species, which became thinner and flatter [54].

Historically, seed mass has been perceived as a trade-off between producing a few large seeds (with a higher probability of survival due to having more food reserves) versus producing numerous small seeds (with a lower survival chance). The amount of energy invested in progeny seed varies between species, especially under stressful conditions [55]. Under stress, the capacity of the progeny seeds to reach optimal energetic investment is limited by the capacity of the plant to access and allocate resources for reproduction [44]. Seed germination and seedling survival have been related to seed size several times in the literature [13]. The fact that *D. brachypodum* did not decrease the seed size under soil moisture stress treatments suggests that this species allocated its resources to reproduction to ensure the fewer seeds that were produced maximized their chances for good seedling establishment.

### 4.3. Post-Anthesis Soil Moisture Effects on Seed and Seedling Traits

No differences in the progeny seed or seedling traits were observed when the maternal plants were exposed to different soil moisture treatments. These findings contrast with other species and those observed for vegetative and phenological plant traits, where significant changes were observed under soil moisture stress. Likewise, no differences in PY (either depth or proportion of seeds presenting it) were observed for different soil moisture treatments, which contrasts with other studies that have shown maternal drought to increase PY in the seeds of Fabaceae species, such as *Glycine max* (L.) (soybean) [56], *A. nitidiflorus* [27], and *Phaseolus vulgaris* L. (common bean) [25]. Although, like our findings, the growth of the Mediterranean plant *Cistus ladanifer* L. (gum rockrose, Cistaceae) was affected by drought conditions while reproductive output, seed size, PY, and viability were unaffected [57].

Soil moisture level has been associated with the seed moisture content reached during seed maturation drying on the mother plant, with seeds that achieve a lower moisture content at dispersal having deeper PY [25]. Seed impermeability development occurs at the end of the maturation drying stage [20] and is only initiated after the seed moisture content falls below a certain threshold level, which is species-specific [25]. The moisture stress treatments tested in this study might not have been sufficiently low to alter this threshold and, therefore, the proportion or depth of PY seeds.

It was observed that stressed plants can produce seeds that are more tolerant to the stress experienced by the maternal plant as compared to the seeds produced under non-stressful environments [58]. Our seeds were sourced from an area (Narrabri, NSW) where drought can be common, but other seed provenances are less likely to occur where moisture stress is more of an issue. Therefore, caution should be taken when extrapolating these findings to *D. brachypodum* seeds from other provenances or other native legume species. Nevertheless, by using a range of temperatures and moisture stress levels, these findings are valuable to enable an expansion in knowledge of plant recruitment from the seeds of *D. brachypodum* under a changing climate. Future research should be undertaken to investigate a wider array of species (and seed provenances) to further build an understanding of how climate change will impact Australia’s native flora. Additionally, the interaction between soil moisture and warming temperatures should be investigated, as warming temperatures can exacerbate soil moisture stress.

## 5. Conclusions

The success of seed-based restoration could be improved if informed approaches are used to source high-quality seeds with adaptive capacities. This is particularly important when restoring highly degraded landscapes threatened by climate change, such as several plant communities around Australia. When planning seed-based restoration projects, we suggest that considering how environmental factors can interact with seed functional traits is crucial to ensure that seeds are sown at the best time and soil depth for seedling establishment. Considering how the maternal environment might have shaped regeneration traits can also have important implications for the success of restored populations. Therefore, when sourcing and selecting legume seeds (such as *D. brachypodum*), having previous knowledge of the temperature and precipitation patterns of the provenance site can be useful in predicting seed quality. In the case of *D. brachypodum*, its reproductive output had low plasticity in response to soil moisture stress, and the seed traits did not vary significantly for the different soil moisture treatments. Therefore, this species would provide more flexibility when sourcing seeds and would be a good candidate when restoring degraded ecosystems susceptible to climate change. However, for other legume species, as shown in the literature, seed traits might significantly vary, impacting the seed quality and restoration outcomes.

The possible decline in seedling establishment during periods of increased temperature and moisture stress, coupled with a possible decline in seed yield during the flowering seasons, may mean certain native species become vulnerable to climate change despite having the capacity to germinate under high soil moisture stress. Consequently, knowledge of the impacts that a changing climate will have on native plant recruitment from seeds is important to understand future plant community structures and, in restoration planning, where an opportunity to select more tolerant species is possible. By deepening our knowledge on these matters, restoration practitioners can have a better understanding of species resilience to different abiotic factors, suitable sources, high-quality seeds, and restoration plans to ensure the newly established plant populations are self-sustaining, functional, and resilient to global changes.

## Figures and Tables

**Figure 1 biology-13-00148-f001:**
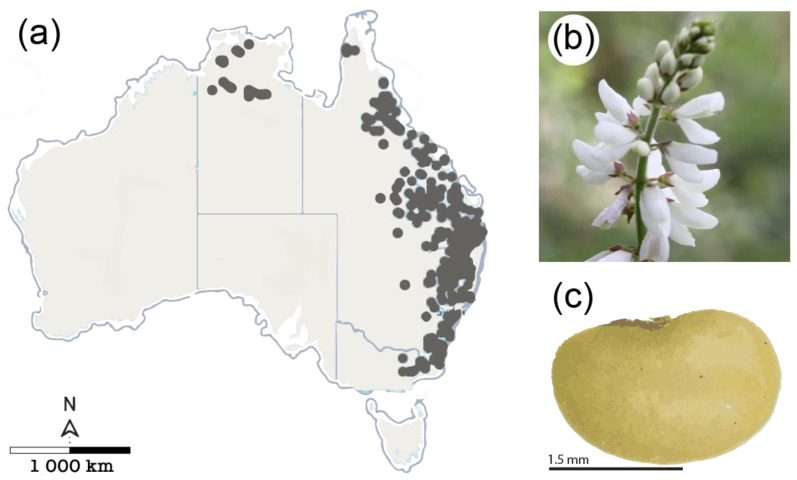
(**a**) Natural distribution of *Desmodium brachypodum* A. Gray in Australia (grey dots), occurring mainly in tropical and subtropical eastern Australia (source: [34]); (**b**) pea-shaped flowers with five petals (two joined) usually paired with a light to dark pink color, occurring in clusters of 8 to 20 flowers; (**c**) small legume seed with a semi-spheroid shape and a smooth, thick yellow testa with a small white elaiosome.

**Figure 2 biology-13-00148-f002:**
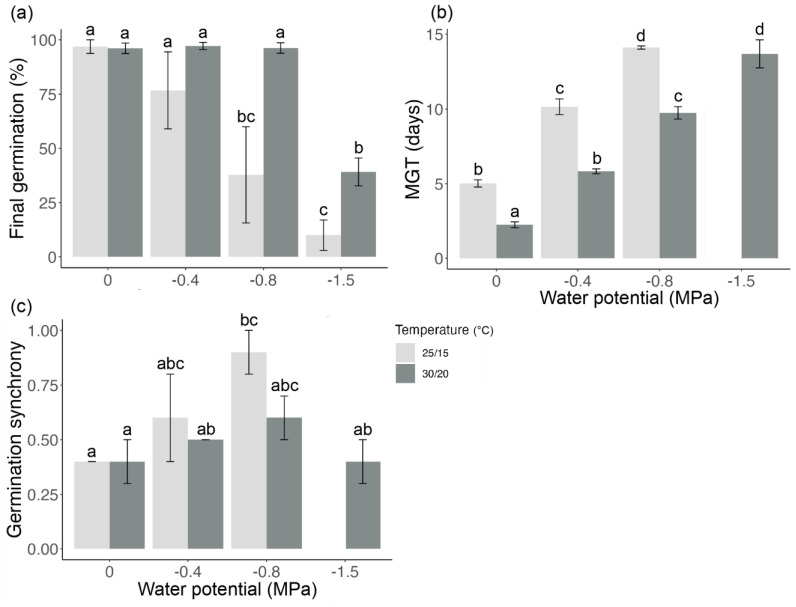
(**a**) Mean seed germination (±SEM), (**b**) mean germination time (MGT), and (**c**) germination synchrony under various temperature and water potential levels (imposed using polyethylene glycol 8000 [PEG] solutions) of the native Australian Fabaceae species *Desmodium brachypodum*. The mean was calculated from four replicates of 20 seeds each. The means followed by a common letter are not significantly different (*p* > 0.05). Given the low germination percentage achieved by the seeds exposed to −1.5 MPa at 25/15 °C, the MGT and germination synchrony were not calculated.

**Figure 3 biology-13-00148-f003:**
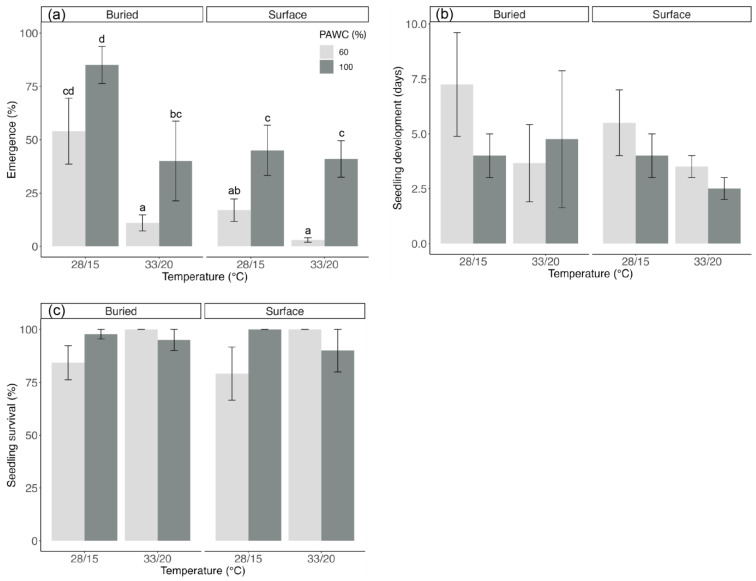
The effects of seed burial, temperature, and plant-available water content (PAWC) on the (**a**) mean final emergence, (**b**) mean seedling development time (days from emergence to first leaf), and (**c**) seedling survival (live seedlings at the end of the trial [60-day trial]—total seedling emergence) ± SEM of the native Australian Fabaceae species *Desmodium brachypodum.* The results show emergence after 60 days after sowing four replicates of 25 seeds each per treatment. The means within the graph (**a**) followed by a common letter are not significantly different (*p* > 0.05); no statistically significant differences occurred for graphs (**b**,**c**).

**Figure 4 biology-13-00148-f004:**
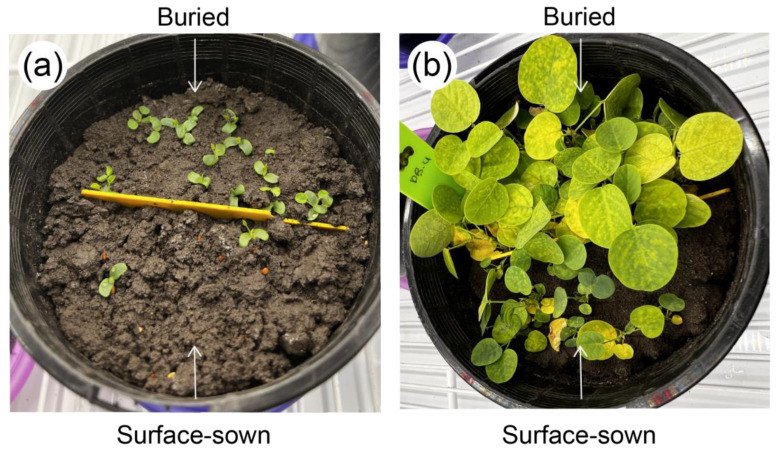
Effects of seed burial on the emergence and establishment of the native legume species *Desmodium brachypodum* under optimal temperature and moisture conditions (28/15 °C and 100% plant-available water content) (**a**) at 6 days after sowing and (**b**) 30 days after sowing.

**Figure 5 biology-13-00148-f005:**
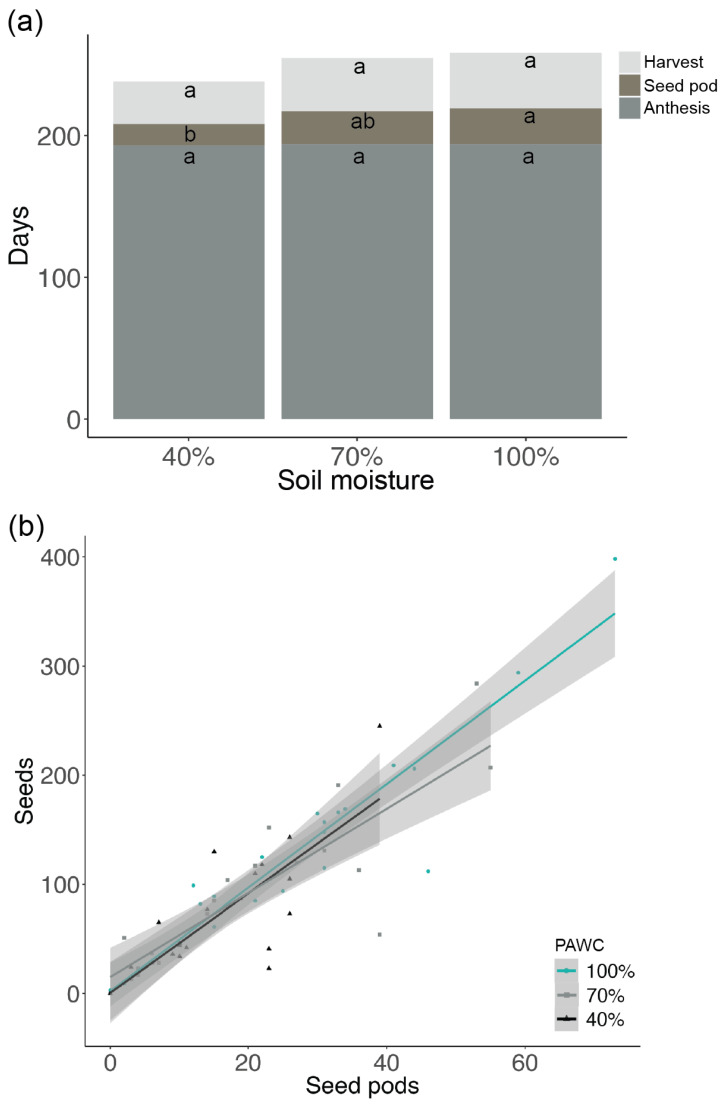
(**a**) Length of the vegetative stages (days from seedling emergence to anthesis) and reproductive stages (days from anthesis to first seed pod formation, and from first seed pod formation until last seed pod harvest) of the native Australian Fabaceae species *Desmodium brachypodum*, exposed to a post-anthesis soil moisture treatment of either 40% plant-available water content (PAWC), 70% PAWC or 100% PAWC. The means followed by a common letter are not significantly different (*p* > 0.05). (**b**) The number of seeds harvested per plant and the number of seed pods produced per plant exposed to different soil moisture treatments.

**Figure 6 biology-13-00148-f006:**
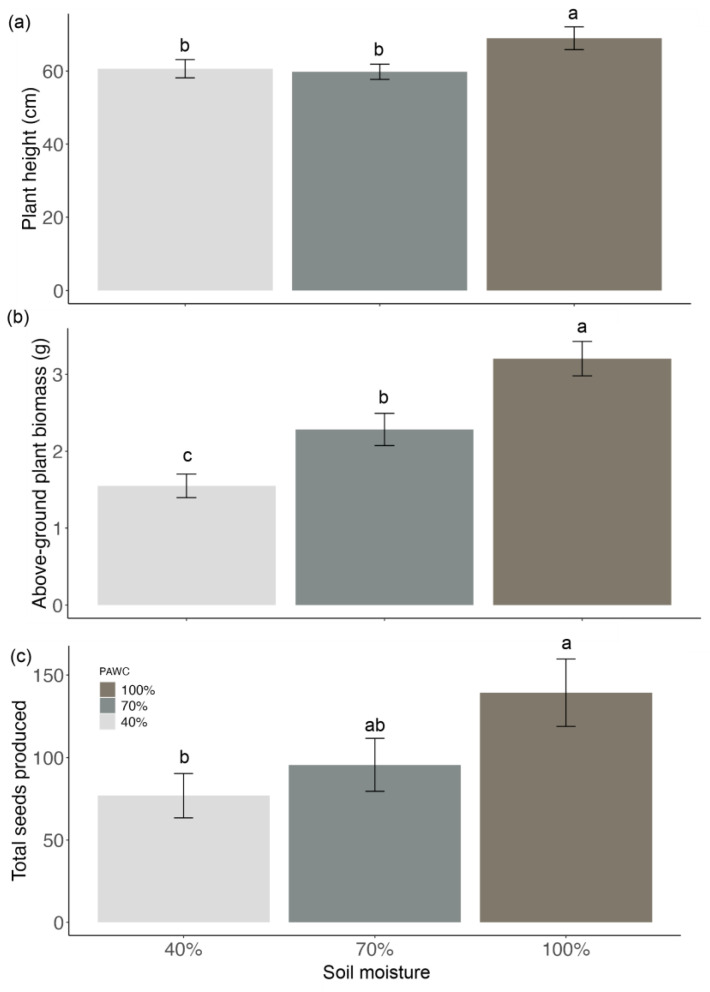
(**a**) Plant height, (**b**) above-ground dry plant biomass, and (**c**) total seeds produced ± SEM of the native Australian Fabaceae species *Desmodium brachypodum.* The plants were exposed to either 40% plant-available water content (PAWC), 70% PAWC, or 100% PAWC. The means within a graph followed by a common letter are not significantly different (*p* > 0.05).

**Figure 7 biology-13-00148-f007:**
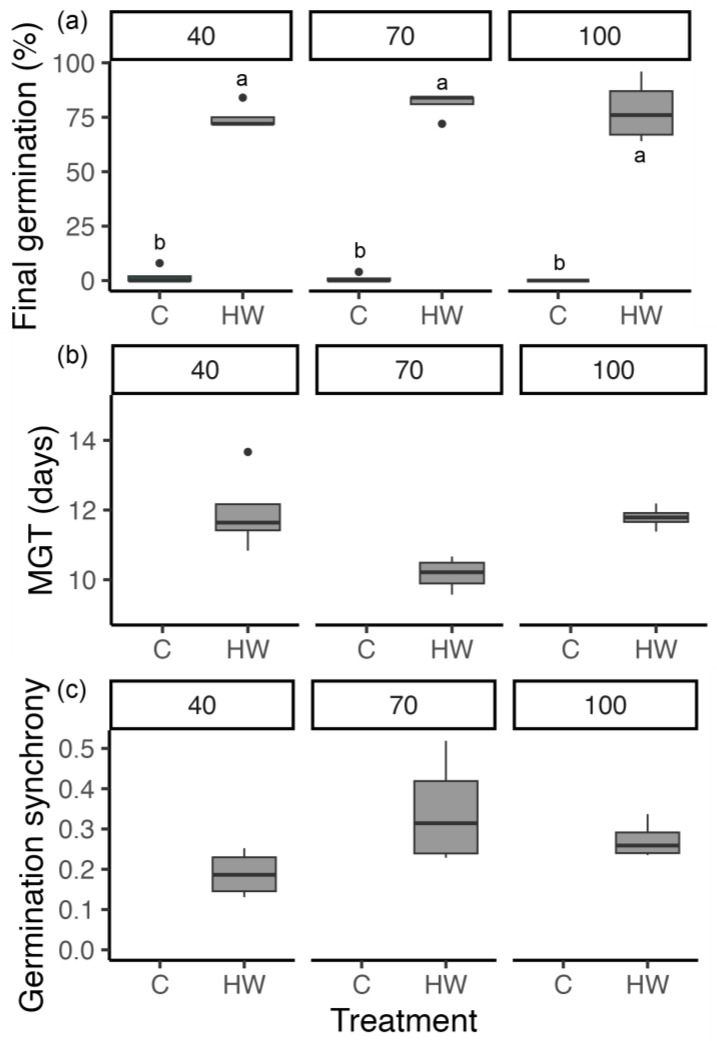
Final seed germination percentage, (**a**) mean germination time (MGT), (**b**) and germination synchrony (**c**) of four replicates (25 seeds each) ± SEM for the progeny seeds of the maternal plants that were exposed to either 40% plant-available water content (PAWC), 70% PAWC, or 100% PAWC from post-anthesis to seed production. Two seed treatments were used: control (C) and hot water (HW). The means within the graph (**a**) followed by a common letter are not significantly different (*p* > 0.05); no statistically significant differences occurred for graphs (**b**,**c**).

## Data Availability

The data are available upon request from the corresponding author.

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
