# Peer review of "Environmental Effects during Early Life-History Stages and Seed Development on Seed Functional Traits of an Australian Native Legume Species"

_biology, 2024, doi:10.3390/biology13030148_

Round 1

Reviewer 1 Report

Comments and Suggestions for Authors

The manuscript “Effects of environmental factors during early life-history and 2 seed development of an Australian native legume species used 3 in seed-based restoration” has been reviewed. The research is presented in clear and well-structured way, relevant to this field of research. Results are well interpreted and understandable followed by consistent conclusions. Minor changes and clarifications are needed.

Author Response

Line 29-33 rephase the section with aims in manner that it do not resembles aims states at the end of the introduction part

Response:

Thanks for all your constructive comments. We rephrased to avoid repetition of same sentences.

Lines 51 – 52 this was stated earlier in lines 48-49, “most vulnerable stage being the transition from germinated seed to emerged seedling”, no need for repetition

Response: Thank you, this has been deleted to avoid repetition.

Lines 125 – 131 text appropriate for introduction

Line 134 please check for coordinate accuracy

Response: coordinates have been corrected.

Lines 112 and 141 Change “gray” to “Gray” after “Desmodium brachypodum A.”

Response: Modified as “Gray”.

Lines 155 – 164 move the equation to Apendix A together with other equations

Response: As this equation is a crucial part of the methods for this experiment and has been usually included in several other previous publications when using PEG solutions to make it easy for the readers, we also decided to keep it in the main text.

Line 174 there is no Appendix 1, only Appendix A, B and C

Response: “1” has been replaced by “A”.

Line 188 there is no Appendix 2, only Appendix A, B and C

Response: “2” has been replaced by “B”.

Line 189 explain the acronym PAWC in the text also not only in appendix and figures

Response: Expanded acronym “plant available water content (PAWC)”

Lines 200-201 Please give precise description that half of pot was with buried seed and half with seed placed on surface

Response: we have edited this sentence for clarity as pointed out, to read the following:Each pot was divided in half, where 25 seeds were sown on the soil surface (in one half of the pot), and 25 seeds were buried to 2 cm depth (in the other half of the pot)”

Lines 203-204 Please give explanation why did you use different temperature settings in Experiment 2 in comparison to Experiment 1

Response: explanation has been included. This is to simulate natural climatic conditions.

Line 211 please explain what is considered as “end of the trial” for seedlings survival because seedlings were periodically removed starting from 30th day post-sowing

Response: end of the trial is 60 days post sowing”, and it has been included in text.

Line 233 there is no Appendix 3, only Appendix A, B and C

Response: “3” has been replaced by “C”.

Line 241 please clarify did the pots received or kept on water content of 287 ml/kg (100% PAWC). Lines 242 explain why the water stress levels are 70% PAWC and 40% PAWC

Response: Plants were ‘kept/maintained’ at 100% PAWC, and 70% and 40% means reduced water supply which is indicated by reducing the water per Kg in text.

Line 248 “(days to anthesis, days to first seed pod, and days to seed maturation)”, explain this section in more detail

Response: each term has been explained accordingly.

Line 262 Seed “fill” or seed “viability”

Response: seed fill

Line 267 Why did you increased number of tested seeds from 20 to 25 compared to experiment 1

Response: Given the accuracy of experimental conditions is slightly less controlled when using soil as compared to Petri dishes, hence we increased the number of seeds sown to account for this.

Line 273 delete one “%”

Response: Deleted

Line 382 Is the “seeds produced” number of seeds?

Response: Yes “the number of” was added for clarity

Line 390 on histogram y axis and in legend, are “total seeds” refer to number of seeds?

Response: yes total seeds is the total number of seeds harvested

Line 559 Do not put quote in conclusion

Response: the citation has been removed

Reviewer 2 Report

Comments and Suggestions for Authors

The MS "Effects of environmental factors during early life-history and seed development of an Australian native legume species used in seed-based restoration" by Beveridge et al. addressed seed-based ecosystem restoration. The results are consistent with the work carried out; the methodology is well explained and referenced so that it can be reproduced. The manuscript is novel and intriguing and may interest Journal readers and other scholars working on this issue worldwide. My comments and suggestions for this manuscript are as follows:

1. Please mention a few examples of ecological restoration case studies with different plant species in the Introduction section.

2. The abstract needs to contain important quantitative data from the experiment. Please mention the future scope of this study. Besides this, a concluding remark is missing in the abstract.

3. To better explore the mechanics underlying this phenomenon and explain the outcomes more thoroughly, kindly include some molecular studies.

4. The conclusion needs to be more thorough. Kindly revise this section to include the important results and insights of the experiments, the underlying mechanisms, and how they support ecological restoration. Additionally, it would be beneficial to discuss any limitations of the study and suggest potential areas for future research to enhance the understanding of ecological restoration practices further. This comprehensive approach will provide a more informative conclusion for readers. 

5. The conclusion implies that Desmodium brachypodum and some other legume crops aid in ecological restoration. How generalizable are these findings to other plant species?

Author Response

Please mention a few examples of ecological restoration case studies with different plant species in the Introduction section.

Response:  Thanks for all the comments. This study is about the interaction effects of environmental conditions on seed traits and mother plant traits. Although the model species is used in seed-based restoration and the outcome would lead to ecological restoration improvements, this study is not about the ecological restoration per se. Hence, authors decided not to deviate the focus of this study, so the additional references were not provided.

The abstract needs to contain important quantitative data from the experiment. Please mention the future scope of this study. Besides this, a concluding remark is missing in the abstract.

Response: To keep the abstract concise, we have stated the most important quantitative information, such as “each by 50%” (Line 35), “by 9 days” (Line 36), and “by almost 50%” (Line 37). The concluding remark at the end of the paragraph has been edited to highlight the concluding remarks.

To better explore the mechanics underlying this phenomenon and explain the outcomes more thoroughly, kindly include some molecular studies.

Response: While molecular studies are an important aspect when studying climate change impacts in plants, this escaped the scope of this manuscript. In this study we focused on how climate change could affect seed biology and ecology of native plant species. Unfortunately, authors think it may deviate from the focus.

The conclusion needs to be more thorough. Kindly revise this section to include the important results and insights of the experiments, the underlying mechanisms, and how they support ecological restoration. Additionally, it would be beneficial to discuss any limitations of the study and suggest potential areas for future research to enhance the understanding of ecological restoration practices further. This comprehensive approach will provide a more informative conclusion for readers. 

Response: Thanks for your comments. Lines 577– 586 explain how the results from this study support ecological restoration. Potential areas for future research are discussed throughout the Discussion section, in Lines 497 – 500, and at the end of the discussion section in Lines 564 – 568. The limitations to this study have been added in Lines 560 – 561.

The conclusion implies that Desmodium brachypodum and some other legume crops aid in ecological restoration. How generalizable are these findings to other plant species?

Response: Although D. brachypodum has been used as a model species, and results from this study are useful to have a general idea of the kinds of impacts climate change will have on native Fabaceae species, caution needs to be taken when generalizing these findings to other species, as the maternal environment can play a crucial role on seed development and seed traits as explained in the discussion, this has been explained in Lines 560 – 561.

Reviewer 3 Report

Comments and Suggestions for Authors

The manuscript by Beveridge and co-authors deals with the prediction of plant establishment under the influence of climate change (air temperature and soil moisture). Based on the analysis of seed germination and seedling establishment under temperature and moisture stress, authors studied how seed burial depth and post-anthesis soil moisture level affect the reproduction of Desmodium brachypodum A. gray. The results obtained are important for understanding the functional trait of seed-based restoration of native legume species. Overall, the scientific background of the manuscript is good and there are only a few minor comments.

1    1) In the title of the manuscript, the authors refer to "Australian native legume species". However, only Desmodium brachypodum A. gray was studied. I would suggest to clarifying the title.

 2)      Line 137: “Average 100-seed weight was 27.0 ± 0.00 g”.    Have you noticed any difference in seed weight?

 3)      Line 173: “Petri dishes were checked for germination three times per week”. I assume you checked the seeds for germination.

 4)      Controlled ageing test (line 268): How did you create relative humidity in the sealed boxes? Did you use a saturated salt solution?

 5)      The duration of the CAT was up to 129 days. We usually have problems with incubation of legume seeds under high temperature and humidity related to the seed rotting.

 6)      Line 407: “CAT test results showed...”. I did not find these results in the Table or Figure.

Author Response

In the title of the manuscript, the authors refer to "Australian native legume species". However, only Desmodium brachypodum A. gray was studied. I would suggest to clarifying the title.

Response: Thanks for all the comments. To clarify, the title states “of an Australian legume”, as we are making reference to one species that was used as a model species. Hence, authors think it is not necessary to rephrase the title.

Line 137: “Average 100-seed weight was 27.0 ± 0.00 g”.    Have you noticed any difference in seed weight?

Response: Yes, we noticed the weight. The differences were too small to be picked up unless we include too many decimal points.

Line 173: “Petri dishes were checked for germination three times per week”. I assume you checked the seeds for germination.

Response: the words “Petri dishes” were changed as “Seeds”.

Controlled ageing test (line 268): How did you create relative humidity in the sealed boxes? Did you use a saturated salt solution?

Response: yes we did. we used the Kew Gardens protocols that use LiCl to control humidity levels. This has been included in the text for clarity in Lines 278 – 279.

The duration of the CAT was up to 129 days. We usually have problems with incubation of legume seeds under high temperature and humidity related to the seed rotting.

Response: Luckily, we did not notice any mould issues during the CAT tests, we kept everything as sterile as possible.

Line 407: “CAT test results showed...”. I did not find these results in the Table or Figure.

Response: As these three P50  values are not extensive data/results to present it in a table we have just written them in the text.